# Immune Checkpoint Inhibitors: A Promising Treatment Option for Metastatic Castration-Resistant Prostate Cancer?

**DOI:** 10.3390/ijms22094712

**Published:** 2021-04-29

**Authors:** Vicenç Ruiz de Porras, Juan Carlos Pardo, Lucia Notario, Olatz Etxaniz, Albert Font

**Affiliations:** 1Germans Trias i Pujol Research Institute (IGTP), Ctra. Can Ruti-Camí de les Escoles s/n, 08916 Badalona, Spain; 2Badalona Applied Research Group in Oncology (B·ARGO), Catalan Institute of Oncology, Ctra. Can Ruti-Camí de les Escoles s/n, 08916 Badalona, Spain; jcpardor@iconcologia.net (J.C.P.); lnotario.germanstrias@gencat.cat (L.N.); oetxaniz@iconcologia.net (O.E.); 3Department of Medical Oncology, Catalan Institute of Oncology, University Hospital Germans Trias i Pujol, Ctra. Can Ruti-Camí de les Escoles s/n, 08916 Badalona, Spain

**Keywords:** metastatic castration-resistant prostate cancer, immunotherapy, immune checkpoint inhibitors, PD-1, PD-L1, CTLA-4

## Abstract

Since 2010, several treatment options have been available for men with metastatic castration-resistant prostate cancer (mCRPC), including immunotherapeutic agents, although the clinical benefit of these agents remains inconclusive in unselected mCRPC patients. In recent years, however, immunotherapy has re-emerged as a promising therapeutic option to stimulate antitumor immunity, particularly with the use of immune checkpoint inhibitors (ICIs), such as PD-1/PD-L1 and CTLA-4 inhibitors. There is increasing evidence that ICIs may be especially beneficial in specific subgroups of patients with high PD-L1 tumor expression, high tumor mutational burden, or tumors with high microsatellite instability/mismatch repair deficiency. If we are to improve the efficacy of ICIs, it is crucial to have a better understanding of the mechanisms of resistance to ICIs and to identify predictive biomarkers to determine which patients are most likely to benefit. This review focuses on the current status of ICIs for the treatment of mCRPC (either as monotherapy or in combination with other drugs), mechanisms of resistance, potential predictive biomarkers, and future challenges in the management of mCRPC.

## 1. Current Treatment Options for Metastatic Castration-Resistant Prostate Cancer (mCRPC)

Prostate cancer (PC) is the most common malignancy in men and a major cause of cancer death [1]. For patients with PC who experience disease relapse after local therapy or for those with metastatic disease, androgen deprivation therapy (ADT) is the backbone of systemic therapy. However, despite significant initial responses to ADT, almost all metastatic patients progress to an incurable metastatic castration-resistant prostate cancer (mCRPC), defined as radiographic progression and/or a rise in prostate-specific antigen (PSA) despite having a castrate level of testosterone [2,3]. Sadly, mCRPC has a poor prognosis with a median overall survival (OS) of approximately 3 years, making it a significant therapeutic challenge. Importantly, the majority of castration-resistant prostate tumors retain their dependence on the androgen receptor (AR), mainly through genomic amplification or gain-of-function mutations of the AR gene, AR mRNA splice variants, or alterations in androgen synthesis [4]. Thus, potent next-generation AR pathway inhibitors (ARPIs), such as abiraterone, an irreversible inhibitor of CYP17A1, and enzalutamide, an AR antagonist, are also commonly used in asymptomatic and mildly symptomatic mCRPC patients [5,6,7,8]. The AR antagonists enzalutamide, apalutamide and darolutamide have also demonstrated improved outcomes in men with non-metastatic CRPC [9,10,11].

Additionally, docetaxel, the standard first-line chemotherapeutic treatment for symptomatic mCRPC patients, has shown a benefit in terms of pain relief, serum PSA levels, quality of life (QoL) and OS [12,13,14]. More recently, docetaxel was also evaluated in high-volume disease metastatic castration-sensitive prostate cancer patients in combination with ADT and positive results were reported in two different clinical trials [15,16]. Nevertheless, as is often the case with chemotherapy, the efficacy of docetaxel is limited by the development of tumor resistance. However, since 2010, men who are resistant to docetaxel-based chemotherapy can receive cabazitaxel, a second-generation taxane, which has shown activity in docetaxel-resistant cell lines [17] and demonstrated clinical efficacy in the pivotal phase III TROPIC [18] and CARD [19] trials.

Unfortunately, very few therapeutic options are available to patients progressing to taxanes, although platinum-based treatments have demonstrated a limited benefit in some patients. Modest response rates have been observed with single-drug platinum-based regimens, but more encouraging results have been obtained with platinum/taxane combinations [20,21]. Of note, Corn and colleagues suggested that cabazitaxel plus carboplatin is well tolerated and confers superior progression-free survival (PFS) and response rate (RR) than cabazitaxel alone in docetaxel-resistant mCRPC patients [22]. Interestingly, patients with aggressive variant prostate cancer (AVPC) [23] were found to benefit most from this combination [22]. AVPC is less dependent on AR signaling and harbors defects in tumor suppressor genes such as *TP53*, *RB1*, and *PTEN* [24,25]. In this context, our group recently demonstrated that the CXCR2/BCL-2 antiapoptotic axis is markedly reduced in taxane-exposed mCRPC tumors, representing an emerging vulnerability to genotoxic platinum-based treatments. Taken together, these findings indicate that taxane-resistant tumors are sensitized to platinum treatment, and taxane/platinum combinations have antitumor efficacy in aggressive mCRPC [26].

Alterations in DNA damage repair (DDR) genes, such as those involving the homologous recombination (HR) pathway genes, such as *BRCA2, BRCA1, ATM, CDK12*, and *PALB2*, occur in approximately 20–25% of mCRPC cases at either the somatic or the germline level [27,28]. In this regard, two Poly (ADP-ribose) polymerase inhibitors (PARPi), rucaparib and olaparib, have been recently approved by the FDA for the treatment of mCRPC [29]. While both agents are approved for tumors with *BRCA1/2* alterations, the approval for olaparib also includes patients with 12 other HR gene alterations, including *ATM* and *PALB2* [30,31]. In fact, the association between alterations in DDR genes and positive response to PARPi is the first example of precision medicine in mCRPC.

In the last years, immunotherapy—particularly immune checkpoint inhibitors (ICIs)—is re-emerging as a viable option for PC, and especially for mCRPC, to stimulate anti-tumor immunity [32,33,34]. Here we review and discuss the rationale, translational background, and clinical trials of ICIs, either as monotherapy or in combination with other drugs currently used in mCRPC.

## 2. ICIs in mCRPC: An Overview

A growing body of evidence has demonstrated that tumors can suppress the antitumor immune response by several mechanisms, including cytokine release, recruitment of immunosuppressive cells, and upregulation of co-inhibitory receptors, known as immune checkpoints [35]. Although immunotherapy has emerged as a promising treatment option for many solid tumors, such as melanoma [36,37], non-small-cell lung cancer (NSCLC) [38], renal cell carcinoma [39], urothelial cancer [40], and head and neck cancer [41], progress in PC has been relatively modest, even though PC was one of the first diseases where an immunotherapeutic agent, sipuleucel-T, was approved. In fact, sipuleucel-T, an autologous antigen presenting cell (APC)-based immunotherapy, has shown a benefit in OS, but not in PFS, in clinical trials [42,43,44] and is currently the only FDA-approved immunotherapy option for mCRPC. 

ICIs are monoclonal antibodies (mAbs) that can inhibit immune checkpoint receptors and therefore prevent the inactivation of T-cell function. The clinically most important immune checkpoint receptors include programmed death 1 (PD-1), programmed death ligand 1 (PD-L1) and cytotoxic T lymphocyte-associated protein 4 (CTLA-4) [32]. Interestingly, mismatch repair deficiency (dMMR) or microsatellite instability-high (MSI-H) tumors are sensitive to standard and novel hormonal therapies and to PD-1 inhibitors [45,46]. In a study by Abida et al., 32 of 1033 (3%) screened CRPC patients had dMMR. Eleven patients were treated with anti-PD-1/PD-L1 therapy; six had a PSA response >50% and four had a radiographic response. Long-lasting disease control was observed in five of six responders [47]. Genomic events leading to dMMR and MSI-H mostly affect the *MSH2* and *MSH6* genes [46,48] and are associated with more aggressive clinical and pathological features and a higher tumor mutational burden (TMB) [49]. Since a high TMB is commonly associated with better clinical outcome to immunotherapy [50,51,52,53], it is important to take into account that PC generally has a lower TMB. Only 3–8.3% of advanced PC tumors have a high TMB, which represents a significant obstacle for immunotherapy efficacy in these patients [54,55]. Here we summarize and discuss the clinical trials evaluating the blockade of the PD-1/PD-L1 and CTLA-4 pathways in mCRPC.

## 3. PD-1/PD-L1 Inhibitors in mCRPC

PD-1 is a transmembrane protein expressed on T cells that interacts with PD-L1, which is expressed on both normal and cancer cells. This interaction is an important checkpoint for CD8+ T-cell inhibition. PD-1 binding to PD-L1 on tumor cells results in an inhibition of apoptosis, T-lymphocyte tolerance, and an increase in tumor cell survival [56]. Currently, ICIs that act as inhibitors of PD-1 include nivolumab and pembrolizumab, whereas anti-PD-L1 mAbs include atezolizumab, avelumab, and durvalumab [56]. Although antibody-mediated blockade of the PD-1/PD-L1 interaction is an effective clinical strategy in several tumor types, data from trials in mCRPC patients suggest that PD-1/PD-L1 blockade is less effective in this disease, with no observable objective responses after nivolumab treatment [57]. In the same line, in a phase Ia study with atezolizumab in 35 mCRPC patients who progressed to sipuleucel-T or enzalutamide, PSA response was only 8.6%, only one patient had an objective partial response, and median OS was 14.7 months [58]. In the KEYNOTE-199 trial, mCRPC patients previously treated with docetaxel and at least one ARPI were treated with pembrolizumab. Three cohorts of patients were included: PD-L1-positive tumors (cohort 1: 133 patients), PD-L1-negative tumors (cohort 2: 66 patients) and bone-predominant disease (cohort 3: 59 patients). The activity of pembrolizumab was poor in all three cohorts, with <10% PSA response and <5% overall response rate (ORR), while radiographic PFS (rPFS) was 2.1, 2.1, and 3.7 months, respectively, in the three cohorts [59].

The value of PD-L1 expression as a prognostic and predictive biomarker of response to ICIs is currently a question for debate due to its variable predictive value among different solid tumor types. Several studies have suggested that the relatively low expression levels of PD-L1 in mCRPC tumors can represent an important obstacle for anti-PD-1/PD-L1 efficacy as monotherapy [60,61]. On that basis, the KEYNOTE-028 trial of pembrolizumab in advanced prostate adenocarcinoma only included pre-treated patients with PD-L1 expression in >1% of tumor or stroma cells. Results showed an ORR of 17.4%, with 34.8% of patients exhibiting stable disease and a median duration of response of 13.5 months after pembrolizumab administration [62].

Interestingly, progression to enzalutamide in mCRPC has been associated with an upregulation of tumor PD-L1, suggesting that PD-L1 might be a dynamic biomarker that is involved not only in immune evasion but also in resistance to enzalutamide [63]. Based on this hypothesis, a promising therapeutic strategy could be the combination of an anti-PD-L1 with enzalutamide. In fact, in a recent study, 28 CRPC patients who had progressed to enzalutamide were treated with pembrolizumab plus enzalutamide. A significant PSA response rate of >50% was observed in 5 (18%) patients; 3 of 12 patients (25%) had a partial response; OS was 21.9 months. Importantly, in two cases, response was maintained after 37 and 39 months [64]. Along the same lines, in cohort C of the KEYNOTE-365 trial, 102 mCRPC patients who had progressed on abiraterone were treated with pembrolizumab plus enzalutamide. PSA response was 22%, ORR was 12%, rPFS was 6.1 months, and OS was 20.4 months [65]. Based on these results, a randomized phase III trial of pembrolizumab plus enzalutamide versus enzalutamide alone is currently ongoing (KEYNOTE-641; NCT03834493). In contrast, in the IMbassador250 phase III trial, the combination of atezolizumab plus enzalutamide did not demonstrate an improvement in OS compared to enzalutamide alone [66].

Taken together, these results suggest that anti-PD1/PD-L1 has a limited activity in mCRPC as monotherapy and combinatory approaches warrant investigation. Several ongoing trials are exploring the promising strategy of PD-1/PD-L1 blockade in combination with other immunotherapies, including vaccines (NCT02933255, NCT03600350, NCT02499835 [67]), kinase inhibitors like ipatasertib (NCT03673787 [68]), ARPIs (NCT02312557, NCT03016312), radium-223 (NCT03093428, NCT02814669 [69]), sipuleucel-T (NCT03024216 [70]), PARPi (NCT03572478; NCT03330405 [71]), navarixin (a CXCR1/CXCR2 antagonist; NCT03473925), and other ICIs, such as the CTLA-4 checkpoint inhibitor ipilimumab (Table 1 and Table 2).

## 4. CTLA-4 Inhibitors in mCRPC

T lymphocytes can express two related transmembrane receptors, CD28 and CTLA-4, on their cell surface, and both bind to the same ligands, CD80 and CD86, on APC. Whereas ligand engagement of CD28 activates T cells, interaction between CTLA-4 and these ligands inhibits T-cell stimulation, thereby promoting a negative feedback loop that prevents T cells from killing cancer cells [56]. Blockade of CTLA-4 is thus an important mechanism for increasing T-cell immunity and, potentially, T-cell antitumor responses. Ipilimumab, a first-in-class fully human mAb that binds to CTLA-4 [81], has been widely studied in several phase I, II, and III clinical trials for mCRPC patients, both as monotherapy and in combination therapy. An initial phase I trial demonstrated that a single dose of ipilimumab given to mCRPC patients was safe and led to a PSA decline of >50% in 14% of patients without inducing significant clinical autoimmunity [82]. On the basis of these results, two phase II trials evaluated the efficacy of ipilimumab in combination with ADT or radiotherapy [77,83]. Both trials demonstrated a greater antitumor activity for ipilimumab plus ADT [83] or radiotherapy [77] as compared to either treatment as monotherapy, with manageable toxicities. Subsequently, preliminary results from two phase III trials demonstrated antitumor activity for ipilimumab, with longer PFS and higher PSA response than placebo, but without significant improvement in OS [74,76]. However, the long-term analysis of the phase III CA184-043 trial showed improved OS with ipilimumab plus radiotherapy compared to placebo plus radiotherapy for post-docetaxel mCRPC patients. Two-, three-, and five-year OS were higher in patients treated with ipilimumab (25% vs 16%, 15% vs 8%, and 7.9% vs 2.7%, respectively) [75,84]. In a recent study of ipilimumab in 30 metastatic PC patients who had an incomplete biochemical response to ADT alone, 30% of patients had a >50% reduction in PSA, with one patient achieving an exceptional >90% reduction. However, this study was interrupted at the interim analysis because the primary endpoint was not achieved [85]. In summary, although these studies suggest that ipilimumab treatment may confer a PFS benefit in mCRPC patients, to date, there is insufficient evidence to justify its use in routine clinical practice.

Previous studies demonstrate that although treatment with ipilimumab increases tumor-infiltrating T cells in PC patients, it also induces compensatory immune-inhibitory pathways, including PD-1/PD-L1, likely suppressing T-cell responses [86]. In fact, due to its poor efficacy as monotherapy, ipilimumab is being tested in several combination trials for mCRPC patients. One promising strategy is combining two different ICIs, as was done in the CheckMate-650 trial, which combined the PD-1 inhibitor nivolumab with ipilimumab in the second- and third-line settings. Results showed a 25% ORR in the pre-chemotherapy cohort 1 and a 10% ORR in the post-chemotherapy cohort 2; median rPFS was 5.5 and 3.8 months, and median OS was 19.0 and 15.2 months, respectively [72]. Interestingly, the biomarker analysis demonstrated improved efficacy in patients with high TMB, HR deficiency, DDR gene alterations, and PD-L1 expression. However, the high rates of toxicity, treatment-related discontinuation, and deaths in the CheckMate-650 trial highlight the need for improving the tolerability of the combination of nivolumab plus ipilimumab [87]. In fact, this study will expand inclusion with 405 additional patients in four arms to assess the role of the combination of ipilimumab and nivolumab at different schedules and in two additional arms with ipilimumab as monotherapy and in combination with cabazitaxel. Similarly, other trials have evaluated the efficacy of ipilimumab in combination with PD-1/PD-L1 inhibitors, especially in patients with AR-V7-positive disease or alterations in DDR genes [73,88].

Finally, ipilimumab has been also tested in combination with ARPIs (NCT01688492), sipuleucel-T [78], vaccines [79], and evofosfamide, an investigational hypoxia-activated prodrug that can potentially reduce hypoxia in the tumor microenvironment (TME) and improve the efficacy of ICIs (NCT03098160) (Table 1 and Table 2).

## 5. Mechanisms of Resistance and Potential Predictive Biomarkers of ICI Response

The real efficacy of ICIs in mCRPC remains elusive and investigational. A better understanding of the mechanisms of sensitivity and resistance to ICIs could help improve the efficacy of these therapies as well as identify new molecular and histological predictive biomarkers [34]. Several potential mechanisms of PC resistance to immunotherapy have been proposed. The slow disease progression of PC has been suggested as a potential explanation for resistance and tolerance to immunotherapy [89]. However, one of the most important factors in tumor resistance to ICIs is the TME. Patients with metastatic PC have dysfunctional cellular immunity and an increased immunosuppressive TME, making the prostate TME unsuitable for tumor-infiltrating immune cells with antitumor activities, such as CD8+ T cells and natural killer (NK) cells, leading to an immunosuppressive environment [90] (Figure 1).

Histocompatibility complex (MHC) Class I proteins are normally expressed on nucleated cells, such as dendritic cells (DC), and present cytosolic peptides to T lymphocytes, triggering an immunostimulatory signal cascade resulting in T cell proliferation and activation [91]. Consequently, loss of MHC Class I expression is a common immune evasion mechanism employed by a variety of cancer types, including PC [91,92]. Interestingly, MHC class I chain-related molecules (MICs), such as MICA and MICB, are proteins expressed in the membrane of tumor cells that bind to the C-type lectin-like stimulatory immune receptor NKG2D in NK cells and CD8+ T cells, activating its cytotoxic effects [93]. Therefore, it has been suggested that the MIC-NKG2D axis participates in epithelial tumor immune surveillance [94]. Of note, it has been shown that aggressive tumors, such as metastatic PC, cleave MICA/B from the membrane of NK and CD8+ T cells, release the soluble form (sMIC) into the plasma, and downregulate the NKG2D receptor from the immune cells, thereby promoting immune suppression and tumor escape [95,96,97]. Several studies in PC have found that the loss of MICA/B expression from the cell surface and the release of sMIC are associated with a more aggressive tumor phenotype [98]. Importantly, high levels of circulating soluble NKG2D ligands have been associated with poor clinical outcome to PD1/PD-L1 blockade [99], while targeting sMIC improves the response of sMIC^+^ tumors to PD1/PD-L1 inhibition by enhancing antigen-specific CD8+ T cell enrichment and function [100].

Furthermore, another factor involved in the importance of the TME in tumor resistance to ICIs is the role of cancer-associated fibroblasts (CAFs), the predominant cell type in the TME. CAFs promote PC carcinogenesis, metastatic progression and therapy resistance by promoting a reactive stroma [101]. It is well described that PC cells establish a crosstalk with CAFs and inflammatory cells to coordinate immune cell recruitment and activation through several signalling factors, including cytokines, chemokines, and growth factors [101,102]. Interestingly, CAFs can recruit monocytes by releasing monocyte chemotactic protein-1 (MCP-1) and stromal cell-derived factor-1 (SDF-1) cytokines, thereby causing them to differentiate into M2-like macrophages, which are capable of exerting their immunosuppressive role via the PD-1 axis. Moreover, it has been suggested that CAFs are able to induce the trans-differentiation of M1 macrophages to tumor-associated M2 macrophages (TAMs) in breast cancer models [103]. In the same line, Ting and colleagues demonstrated that the inhibition of CAF-mediated MCP-1 secretion reduces PC tumor growth and in turn is associated with a reduction in immune cell recruitment [104]. Taken together, these studies suggest that targeting CAFs may represent a new attractive therapeutic strategy to use in combination with ICIs. CAFs have a high expression of fibroblast activation protein-alpha (FAPα), which has been shown to be involved in resistance to immunotherapy [105]. However, although FAP depletion synergizes with anti-PD-L1 immunotherapy in preclinical models of pancreatic ductal carcinoma [106], sadly, FAP-targeted approaches have shown a lack of efficacy in clinical trials [107]. However, novel and potent antibody–drug conjugates like OMTX705 have shown preclinical antitumor activity as single agents and, in combination with chemotherapy in tumors resistant to PD-1 inhibitors, suggest that they may represent an important therapeutic target in combination with ICIs in mCRPC [108].

On the other hand, mCRPC patients present an increased number of immunosuppressive cells including myeloid-derived suppressor cells (MSDC) and regulatory T (Treg) cells in the TME and in peripheral blood [109,110]. Treg cells, one of the key components in tumor immune tolerance and evasion, act by suppressing the pro-inflammatory type 1 CD4+ helper T (Th1) and CD8+ T cells [111]. Cytokines, such as vascular endothelial growth factor (VEGF), transforming growth factor β (TGF-β), interleukin-10 (IL-10), and prostaglandin E2, in the TME are responsible for the recruitment of Tregs and the inhibition of proliferation, activation, and infiltration of cytotoxic lymphocytes [112]. Therefore, depletion of Treg cells is being studied as a possible cancer therapy that could improve the therapeutic effect of PD-1/PD-L1 and CTLA-4 inhibitors [113]. Interestingly, Jiao and colleagues demonstrated that mCRPC bone metastases promote osteoclast-mediated bone resorption that releases TGF-β, which restrains Th1 lineage development. Therefore, blocking TGF-β along with ICIs increases Th1 subsets, promotes clonal expansion of CD8+ T cells and subsequent regression of bone metastases, and improves survival [114]. In addition to the presence of immunosuppressive Treg cells, several reports have demonstrated that high infiltration of M2 TAMs in the PC TME is pro-tumorigenic. In fact, these macrophages secrete high levels of M2-associated immunosuppressive cytokines and chemokines, with TGF-β2 being the most highly expressed [115]. Given the established role of TGF-β in immune evasion, this may be an important factor contributing to poor infiltration of cytotoxic lymphocytes in PC. These findings reinforce the critical immunosuppressive role of TGF-β in the context of current ICIs, where targeting TGF-β prior to ICI treatment has been suggested as an approach to improve response [116]. Interestingly, in preclinical models, PTEN-null prostate tumors are strongly infiltrated by TAMs expressing CXCR2, and activation of this receptor through CXCL2 polarizes macrophages toward an anti-inflammatory phenotype. Pharmacological blockade of the CXCR2 receptor by a selective antagonist promoted the re-education of TAMs toward a pro-inflammatory phenotype [117]. In fact, a phase II trial combining pembrolizumab with the CXCR1/CXCR2 antagonist navarixin in mCRPC patients is currently ongoing (NCT03473925). Additionally, Wise and colleagues described Dickkopf-1 (DKK1) gene expression as a novel contributor to the immunosuppressive TME of mCRPCs with low AR and without neuroendocrine signaling [118]. These data provided the rationale for an ongoing clinical trial targeting DKK1 in mCRPC (NCT03837353). In conclusion, the development of mechanism-driven immunotherapies that can restore NK and CD8+ T-cell function, as well as overcome the immunosuppressive prostate TME, are essential to improving the efficacy of ICIs in mCRPC.

Although this remains a controversial issue, compelling evidence indicates that tumors with an elevated TMB, such as melanoma or NSCLC, promote the expression of mutation-associated neoantigens that might be recognized by T cells, which then attack tumor cells [52,53]. PC has what is commonly referred to as a “cold” TME, with a lower level of tumor-associated antigens and neoantigens, which is an important mechanism of resistance to ICIs [55,119]. Interestingly, defects in HR pathway genes may also be associated with an increase in genomic instability, neoantigen load, PD-L1 expression, and tumor-infiltrating lymphocytes [120]. Alterations in HR pathway genes, including *BRCA2*, *BRCA1*, *ATM*, *CDK12*, and *PALB2*, occur in approximately 20–25% of mCRPC cases at either the somatic or the germline level [27,28], making these patients potentially more responsive to ICIs and suggesting that HR pathway genes may be potential predictive biomarkers of response to ICIs. Indeed, several studies, including the KEYNOTE-199 and CheckMate-650 trials, have shown a trend towards a higher ORR to pembrolizumab in HR-deficient mCRPC patients [59,72]. Particularly, mCRPC tumors with *CDK12* biallelic mutations (approximately 4–7% of all mCRPC tumors) have a unique immune signature and a large number of gene fusions, with an increase in neoantigens and subsequent T-cell infiltration, indicating that these *CDK12* mutations may be a useful predictive biomarker to identify mCRPC patients likely to respond to PD-1 inhibitors [121,122]. Clinical trials to prospectively assess the efficacy of ICIs in patients with *CDK12* alterations are ongoing (NCT03810105, NCT03570619). Interestingly, preliminary evidence also suggests that AR-V7-expressing metastatic PC, an aggressive tumor phenotype with poor PFS and OS, may be enriched for DNA-repair defects, rendering them more sensitive to immune-checkpoint blockade [123] (NCT03061539, NCT02601014; Table 1 and Table 2). However, a recent phase II non-randomized trial showed only modest activity for nivolumab plus ipilimumab in patients with AR-V7-expressing PC [73].

The role of alterations in DDR genes in PARPi sensitivity is widely known [30,31] and several trials have evaluated the combination of ICIs with PARPi. The efficacy of pembrolizumab plus olaparib was evaluated in the cohort A of the KEYNOTE-365 trial, with promising results. Seven patients (9%) achieved a PSA response of >50%, eight (10%) had an ORR, and median OS was 14 months [80]. In addition, another trial combining the PARPi rucaparib with nivolumab is currently ongoing (NCT03572478; Table 2).

Another potential biomarker of sensitivity to ICIs may be PD-L1 expression. In fact, PD-L1 is downregulated in many advanced PC cases, which might partly explain the negative results observed in trials with PD-1/PD-L1 inhibitors [60,61,124]. It has been suggested that PD-L1 expression varies in the different stages of PC progression and may depend on therapies received prior to disease progression [125]. Although the KEYNOTE-028 trial suggested that PD-L1 expression could predict response to ICIs [62], the larger KEYNOTE-199 trial found that the ORR in the PD-L1-negative and the PD-L1-positive cohorts was 3% and 5%, respectively [59]. These results illustrate the low accuracy of PD-L1 as a biomarker of response to ICIs in mCRPC patients. Moreover, it is also important to take into account that PD-L1 expression—in PC and in other cancers—may vary depending on the assay technique and antibody clone as well as on the cut-off used to define positivity/negativity. The fact that the method used for assessing PD-L1 expression is not yet standardized is an important barrier to its clinical use as a predictive biomarker.

In conclusion, although recent years have witnessed some progress, further research is warranted to elucidate the biological and pathological mechanisms underlying resistance to ICIs in order to find effective predictive biomarkers of response.

## 6. Conclusions and Future Challenges

Taken together, all these data suggest that while it is true that one of the most promising approaches to activate therapeutic antitumor immunity is the blockade of immune checkpoints such as CTLA-4 or PD-1/PD-L1, the overall activity of immune checkpoint mAbs in unselected patients with advanced PC has so far been limited. However, it is noteworthy that some subgroups of PC patients may obtain a long-term benefit from immunotherapy; therefore, it is important to focus our efforts on elucidating the biological and pathological mechanisms underlying resistance to ICIs to identify those patients who are likely to benefit. We must be able to extrapolate and adapt the knowledge acquired in other tumors, where immunotherapy is more established, and apply it in PC. To date, there are several biomarkers being studied that may help to predict the benefit of CTLA-4 and PD-1/PD-L1 inhibition, including PD-L1 expression in tumor and immune cells and high TMB in patients with dMMR, MSI-H or HR deficiency. However, we must be aware that it is unlikely that a single marker will be able to predict response to immunotherapy and a panel of biomarkers will probably be needed to define which patients may benefit from these treatments.

In fact, there are currently over 100 ongoing immunotherapy-based clinical trials in PC, many of which are testing combination therapies or new ICIs. For instance, B7-H3, a member of the B7 ligands superfamily, is a newly investigated immune checkpoint target. Although its receptor is still unknown, B7-H3 is expressed by prostate tumor cells [126,127] and its overexpression inhibits T-cell function, thereby contributing to immune evasion [128,129]. For this reason, B7-H3 inhibitors, such as enoblituzumab, are under clinical investigation in several phase I and II trials in metastatic PC, with promising preliminary results [130] (NCT02628535, NCT02923180, NCT01391143). In addition, other immune checkpoint targets are in various stages of clinical development, including LAG-3 [131], OX40 [132], and 4-1BBL [133].

In summary, although there are still many obstacles to overcome, the scenario of immunotherapy in PC is encouraging. Current advances in the combination of immunotherapy with other treatments, the identification of predictive biomarkers, and the determination of new immune checkpoint targets suggest that immunotherapy will be a promising strategy in mCRPC in the coming years.

## Figures and Tables

**Figure 1 ijms-22-04712-f001:**
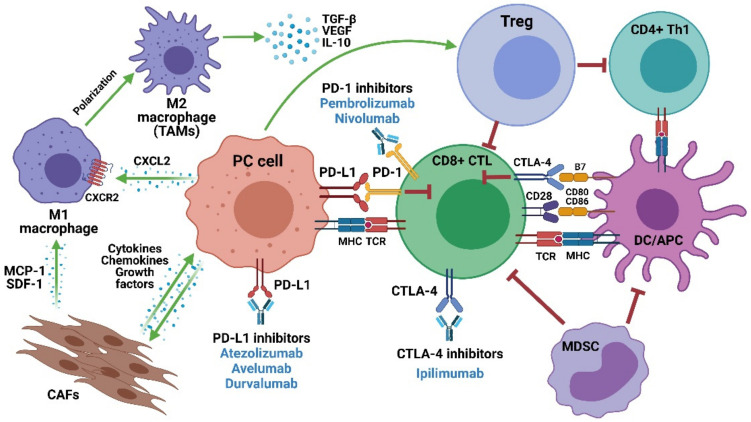
An overview of immune cell types involved in the PC TME and the molecular mechanisms of PD-1 and CTLA-4 attenuation of CD8+ T cell activation. APC: antigen-presenting cell; CAFs: cancer-associated fibroblasts; CD8 + CTL: CD8+ cytotoxic T lymphocyte; CD4 + Th1: type 1 CD4+ helper T; CTLA-4: cytotoxic T lymphocyte-associated protein 4; DC: dendritic cell; MDSC: myeloid-derived suppressor cell; MHC: histocompatibility complex; PC: prostate cancer; PD-L1: programmed death ligand 1; PD-1: programmed death 1; Treg: regulatory T cells; TAMs: tumor-associated macrophages; TCR: T cell receptor. Created with BioRender.com (accessed on 29 April 2021).

**Table 1 ijms-22-04712-t001:** Summary of completed clinical trials evaluating the efficacy of ICIs in monotherapy and in combination with other treatments in mCRPC. **AE:** adverse events; **MTD:** maximum tolerated dose; **ORR:** overall response rate; **OS:** overall survival; **PFS:** progression-free survival; **rPFS**: radiographic progression-free survival; **RP:** radiographic progression.

Trial ID	Treatment	Patients	Primary End Point	Patients	Trial Phase	Results
NCT02787005 (KEYNOTE-199) [59]	Pembrolizumab	Chemotherapy-resistant mCRPC	ORR	370	II	Substantial antitumor activity with an acceptable safety profile
NCT02054806 (KEYNOTE-028) [62]	Pembrolizumab	mCRPC with PD-L1 expression in >1% of tumor or stromal cells	ORR	23	Ib	ORR: 13%
NCT02312557 [64]	Pembrolizumab + Enzalutamide	Enzalutamide resistant mCRPC	PSA response	20	II	PSA response: 30%
NCT02499835 [67]	Pembrolizumab + pTVG-HP(DNA vaccine)	Hormone-resistant mCRPC	AE	32	I/II	Acceptable safety profile
			PFS			
			RP			
			ORR			
			PSA response			
NCT00730639 (MDX-1106)[57]	Nivolumab	mCRPC	AE	395	Ib	No favorable ORR
NCT02985957 (CheckMate-650) [72]	Nivolumab + Ipilimumab	mCRPC	ORRr PFS	497	II	Superior ORR (26%) in chemotherapy-naïve patients
NCT02601014 (STARVE-PC)[73]	Nivolumab + Ipilimumab	mCRPC with detectable AR-V7	PSA Response Safety	15	II	Favorable outcomes in patients with AR-V7 + PC with DDR
NCT00861614 (CA184-043) [74,75]	Ipilimumab	mCRPC following docetaxel therapy	OS	988	III	No significant improvement in OS Increased PFS and PSA response Long-term analysis: OS improvement in Ipilimumab arm
NCT01057810 (CA184-095) [76]	Ipilimumab	Chemotherapy-naïve mCRPC	OS	837	III	No significant improvement in OS Increased PFS and PSA response
NCT00323882 [77]	Ipilimumab + Radiotherapy	mCRPC	AE PSA response Tumor response	75	I/II	Manageable AEs and PSA responses suggestive of clinical activity
NCT01832870 (SIPIPI) [78]	Ipilimumab + Sipuleucel T	Progressive mCRPC	Antigen-specific memory T-cell response	9	I	Acceptable safety profile
NCT01510288 [79]	Ipilimumab + GVAX	mCRPC	AE	28	I	Acceptable safety profile
NCT03016312 (IMbassador250) [66]	Atezolizumab + Enzalutamide vs. Enzalutamide	mCRPC	OS	730	III	Atezolizumab + enzalutamide do not show improvement in OS over enzalutamide alone

**Table 2 ijms-22-04712-t002:** Summary of ongoing clinical trials evaluating the efficacy of ICIs in monotherapy and in combination with other treatments in mCRPC. **AE:** adverse events; **DLT:** dose limiting toxicities; **MTD:** maximum tolerated dose; **ORR:** overall response rate; **OS:** overall survival; **PFS:** progression-free survival; **rPFS**: radiographic progression-free survival; **RP:** radiographic progression.

Trial ID	Treatment	Indication	Primary End Point	Patients	Trial Phase	Preliminary Results
NCT02861573 (KEYNOTE-365) [65,80]	Pembrolizumab + Coh. A: Olaparib Coh. B: Docteaxel Coh. C: Enzalutamide	Abiraterone resistant mCRPC	ORR PSA Response Safety	210	Ib/II	Cohort A:
						ORR: 8%
						PSA Resp: 9%
						Cohort B:
						ORR: 23%
						PSA Resp: 34%
						Cohort C:
						ORR: 12%
						PSA Resp: 22%
NCT03834493 (KEYNOTE-641)	Pembrolizumab + Enzalutamide	mCRPC	OS rPFS	1200	III	N/A
NCT03093428	Pembrolizumab + Radium-223	mCRPC	Extent Of Immune Cell Infiltration	45	II	N/A
NCT03473925	Pembrolizumab + Navarixin	mCRPC	ORR	120	II	N/A
NCT03040791 (ImmunoProst)	Nivolumab	mCRPC with DNA repair defects	PSA Response	45	II	N/A
NCT03572478	Nivolumab + Rucaparib (PARPi)	mCRPC	DLT T Cell Inflammation	12	I/II	N/A
NCT03061539 (NEPTUNES)	Nivolumab + Ipilimumab	mCRPC with specific immunogenic signatures	RR PSA response CTCs	175	II	N/A
NCT03098160	Ipilimumab + Evofosfamide	Metastatic PC	MTD	69	I	N/A
NCT03024216 [70]	Atezolizumab + Sipuleucel-T	Asymptomatic or minimally symptomatic mCRPC	AE	37	Ib	Manageable safety profile
NCT03673787 [68]	Atezolizumab + Ipatasertib	mCRPC with PTEN loss	MTD AE	51	I/II	Well-tolerated
NCT02814669 [69]	Atezolizumab + Radium-223	ARPI-resistant mCRPC	Safety ORR	45	I	No dose-limiting toxicities, safety signals, or changes in serum biomarkers
NCT03330405 (JAVELIN PARP Medley) [71]	Avelumab + Talazoparib	mCRPC	DLT ORR	216	Ib/II	Preliminary antitumor activity and manageable safety profile
NCT03204812	Durvalumab + Tremelimumab	Chemotherapy naïve CRPC	AE	27	II	N/A

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
