# Peer review of "Immune Checkpoint Inhibitors: A Promising Treatment Option for Metastatic Castration-Resistant Prostate Cancer?"

_ijms, 2021, doi:10.3390/ijms22094712_

Round 1

Reviewer 1 Report

The review presented by Dr de Porras and colleagues is focused on the importance of the immune-checkpoint and its inhibition in Prostate Cancer.

In particular, they investigate the role of PD-1/PD-L1  and CTLA-4 inhibitors in Castration-Resistant Prostate Cancer (CRPC).

The topic is very intersting. The English style should be only more fluent and an image should be appreciated.

I have only some curiosity:

-In the field of Prostate Cancer (PC), some emerging data show the importance of carcinoma-associated fibroblasts (CAFs), also when prostate cancer epithelial cells are not dependent on androgens, such as PC3 cells that are androgen receptor (AR) negative. At this point are there studies that correlate the immune-checkpoint with PC tumor microenvironment or in particular with CAFs?

-There are also other important proteins involved in immune-checkpoint, such as for example MICA/B. Are there data about the involvement of MICA/B in PC tunorigenesis?

Reviewer 2 Report

This review article is centred on immune checkpoint inhibitors in the possible treatment for metastatic castration-resistant prostate cancer. Different aspects are discussed starting from the current state of play of ICI treatment up to its future challenges, describing on its way mechanisms of resistance and potential predicative biomarkers. The results from different clinical trials are discussed

In my opinion, the review is interesting and well written covering an important topic.

The clinical trials, both past and present are correctly organized in Tables 1 and 2, however, I miss the use of figures and schemes to summarize the mechanisms of action of the different treatments.

It would be helpful to the reader if the authors include a list of abbreviations and glossary of terms.

I am not sure about the question mark in the title. For me it means the authors doubt the validity of ICI treatment which puts the review and its content, to use a footballing term, offside. However, in the conclusion the authors say there is potential for ICI combination therapies (which I agree with). The authors should reconsider the title.

Reviewer 3 Report

Dear Editor, thank you so much for inviting me to revise this manuscript about immunotherapy in prostate cancer.

Prostate cancer (PC) is the most commonly diagnosed cancer in men, representing one of the leading causes of cancer-related death worldwide. Although patients with localized disease are typically treated with definitive therapy (prostatectomy or radiotherapy or both), up to 40% of subjects receiving radical prostatectomy (RP) and up to 50% of patients receiving radiotherapy will experience recurrence of disease. Consequently, many patients affected by metastatic disease will develop metastatic castration resistant PC (mCRPC). Due to the improved knowledge in terms of molecular mechanisms underlying progressive disease and metastatic onset, in the past two decades we have witnessed a considerable increase in the number of therapeutic options for mCRPC, with several agents entered into everyday clinical practice, including docetaxel, cabazitaxel, abiraterone, enzalutamide and radium-223.

Unfortunately, immune checkpoint inhibitors (ICIs) are not included among these drugs. In fact, although modern immunotherapy has revolutionized the management of a number of malignancies, ICIs are still looking for their niche in several tumors where these agents do not seem to provide ideal results in unselected patients - such as mCRPC. Despite FDA in 2010 approved sipuleucel-T , thus becoming the first immunotherapy for mCRPC, recent trials assessing the role of ICIs have shown disappointing results; so far, the only approved ICI is the anti-PD-1 agent pembrolizumab, which can be used in the subgroup of mCRPC patients with high microsatellite instability (MSI-H). If PD-L1 status, MSI-H and other biomarkers may identify a subset of patients who are most likely to respond, improving the precision in order to select the responders is a major goal.

Based on these premises, this study addresses a current topic.

The manuscript is quite well written and organized. English could be improved.

The tables are comprehensive and clear.

The introduction explains in a clear and coherent manner the background of this study.

We suggest the following modifications:

  • Introduction section: although the authors correctly included important papers in this setting, we believe a couple of studies should be cited within the introduction (PMID: 28663235 ; PMID: 32911806) only for a matter of consistency. We think it might be useful to introduce the topic of this interesting study.
  • The authors should expand the section regarding predictive biomarkers of response to immunotherapy, including a more personal perspective to reflect on. For example, they could answer the following questions – in order to facilitate the understanding of this complex topic to readers: What are the knowledge gaps and how do researchers tackle them? How do you see this area unfolding in the next 5 years? We think it would be extremely interesting for the readers.

However, we think the authors should be acknowledged for their work. In fact, they correctly addressed an important topic in prostate cancer, the methods sound good and their discussion is well balanced.

We believe this article is suitable for publication in the journal although major revisions are needed. The main strengths of this paper are that it addresses an interesting and very timely question and provides a clear answer, with some limitations.

We suggest a linguistic revision and the addition of some references for a matter of consistency. Moreover, the authors should better clarify some points.

Round 2

Reviewer 3 Report

The authors extensively modified the paper according to our suggestions.

We recommend Acceptance in its current form.

Author Response

We are very grateful to the reviewer for his/her appreciation of our work